



# Wave propagation in the Lorenz-96 model

Dirk L. van Kekem[1] and Alef E. Sterk[1]

[1]Johann Bernoulli Institute for Mathematics and Computer Science, University of Groningen, PO Box 407, 9700 AK Groningen, The Netherlands

*Correspondence to:* Alef E. Sterk (a.e.sterk@rug.nl)

**Abstract.** In this paper we study the spatiotemporal properties of waves in the Lorenz-96 model and their dependence on the dimension parameter $n$ and the forcing parameter $F$. For $F > 0$ the first bifurcation is either a supercritical Hopf or a double-Hopf bifurcation and the periodic attractor born at these bifurcations represents a traveling wave. Its spatial wave number increases linearly with $n$, but its period tends to a finite limit as $n \to \infty$. For $F < 0$ and odd $n$ the first bifurcation is again

a supercritical Hopf bifurcation, but in this case the period of the traveling wave also grows linearly with $n$. For $F < 0$ and even $n$, however, a Hopf bifurcation is preceded by either one or two pitchfork bifurcations, where the number of the latter bifurcations depends on whether $n$ has remainder 2 or 0 upon division by 4. This bifurcation sequence leads to standing waves and their spatiotemporal properties also depend on the remainder after dividing $n$ by 4. Finally, we explain how the double-Hopf bifurcation can generate two or more stable waves with different spatiotemporal properties that coexist for the same

parameter values $n$ and $F$.

*Copyright statement.* TEXT

## 1   Introduction

In this paper we study the Lorenz-96 model which is defined by the equations

$$\frac{dx_j}{dt} = x_{j-1}(x_{j+1} - x_{j-2}) - x_j + F, \quad j = 0, \ldots, n-1, \tag{1}$$

together with the periodic "boundary condition" implied by taking the indices $j$ modulo $n$. The dimension $n \in \mathbb{N}$ and the forcing parameter $F \in \mathbb{R}$ are free parameters. The model was introduced by Lorenz (2006) for numerical experiments in predictability studies. In his paper, Lorenz interpreted the variables $x_j$ as values of some atmospheric quantity in $n$ equispaced sectors of a latitude circle, where the index $j$ plays the role of "longitude". Lorenz also remarked that the vectors $(x_0, \ldots, x_{n-1})$ can be interpreted as wave profiles, and he observed that for $F > 0$ sufficiently large these waves slowly propagate "westward", i.e.

in the direction of decreasing $j$. Figure 1 shows a Hovmöller diagram illustrating two traveling waves with wave number 5 for dimension $n = 24$ and the parameter values $F = 2.75$ (in the periodic regime) and $F = 3.85$ (in the chaotic regime).

Table 1 lists some recent papers with applications of the Lorenz-96 model. In most studies the dimension $n$ is chosen ad hoc, but $n = 36$ and $n = 40$ appear to be popular choices. Many applications are related to geophysical problems, but the model



has also attracted the attention of mathematicians working in the area of dynamical systems for phenomenological studies in high-dimensional chaos. Note that Eq. (1) is in fact a family of models parameterized by means of the discrete parameter $n$. An important question is to what extent both the qualitative and quantitative dynamical properties of Eq. (1) depend on $n$. Answers to these questions can be used in selecting the most appropriate values of $n$ and $F$ in specific applications. For example, the

statistics and predictability of extreme events can depend very much on the dynamical regime of a model (Holland et al., 2012; Sterk and Van Kekem, 2017).

In this paper we address the question how the spatiotemporal properties of waves, such as their period and wave number, in the Lorenz-96 model depend on the dimension $n$ and whether these properties tend to a finite limit as $n \to \infty$. We will approach this question by studying waves represented by periodic attractors that arise through a Hopf bifurcation of a stable

equilibrium. Along various routes to chaos these periodic attractors can bifurcate into chaotic attractors representing irregular waves which "inherit" their spatiotemporal properties from the periodic attractor. For example, the wave shown in the left panel of Fig. 1 bifurcates into a 3-torus attractor which breaks down and gives rise to the wave in the right panel. Note that both waves have the same wave number. Figure 2 shows power spectra of these waves, and clearly their dominant peaks are located at roughly the same period. Inheritance of spatiotemporal properties also manifests itself in a shallow water model

studied by Sterk et al. (2010) in which a Hopf bifurcation (related to baroclinic instability) explains the observed time scales of atmospheric low-frequency variability.

In addition to *traveling* waves, such as illustrated in Fig. 1, we will also show the existence of *standing* waves. In a recent paper by Frank et al. (2014) standing waves have also been discovered in specific regions of the *multi-scale* Lorenz-96 model. Their paper uses dynamical indicators such as the Lyapunov dimension to identify the parameter regimes with standing waves.

Moreover, we will explain two bifurcation scenarios by which waves with different spatiotemporal properties coexist. This paper complements the results of our previous work (Van Kekem and Sterk, 2017a) which restricted to the classical case $F > 0$, and thereby we give a comprehensive picture of wave propagation in the Lorenz-96 model.

The remainder of this paper is organized as follows. In Sect. 2 we explain how to obtain an approximation of the periodic attractor born at a Hopf bifurcation which allows us to derive spatiotemporal properties of the waves in the Lorenz-96 model. In

Sect. 3.1 we show that for $F > 0$ periodic attractors indeed represent traveling waves as suggested by Lorenz. Also for $F < 0$ and odd values of $n$ periodic attractors represent traveling waves, as is demonstrated in Sect. 3.2. In Sect. 3.3, however, we show analytically that for $n = 6$ and $F < 0$ *standing* waves occur. By means of numerical experiments we show in Sect. 3.4 that standing waves occur in general for even $n$ and $F < 0$. In Sect. 4 we discuss the bifurcation scenarios by which stable waves with different spatiotemporal properties can coexist for the same values of the parameter $F$.

## 2   Hopf bifurcations

In this section we consider a general geophysical model in the form of a system of ordinary differential equations:

$$\frac{d\boldsymbol{x}}{dt} = \boldsymbol{f}(\boldsymbol{x}, \mu), \qquad \boldsymbol{x} \in \mathbb{R}^n. \tag{2}$$



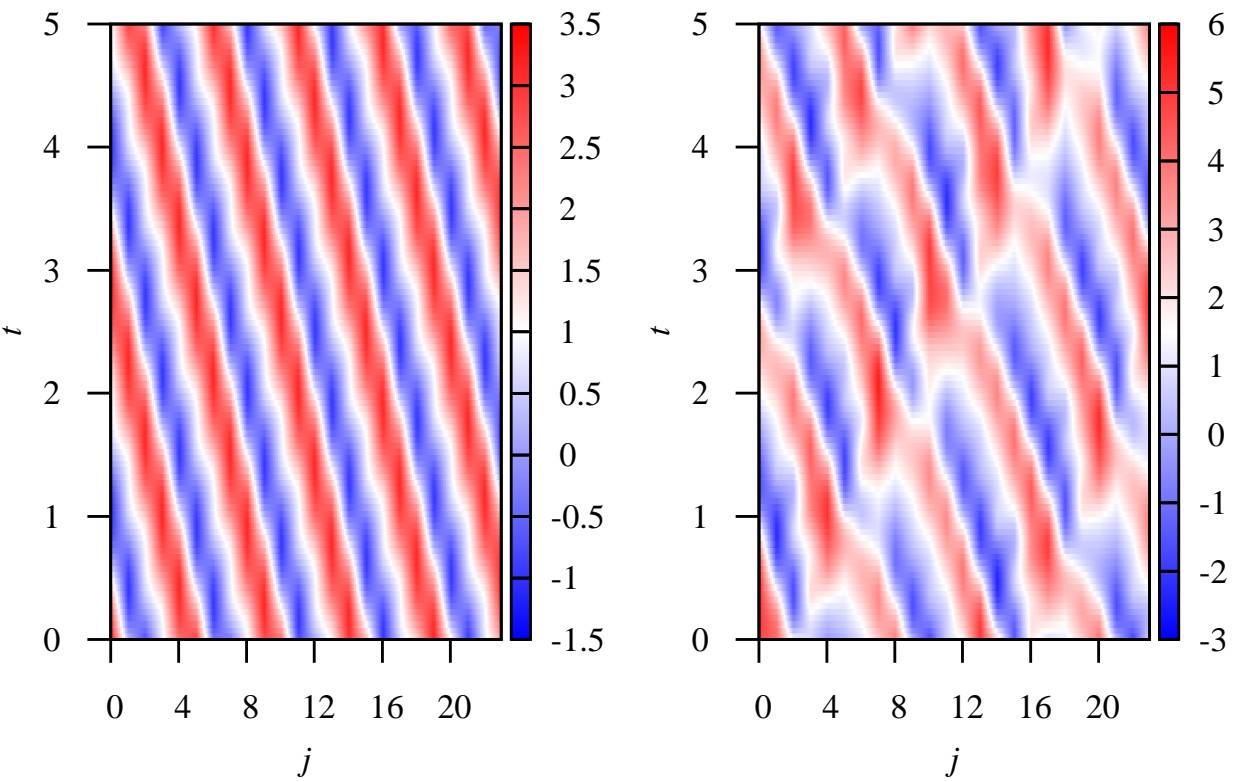

**Figure 1.** Hovmöller diagrams of a periodic attractor (left, $F = 2.75$) and a chaotic attractor (right, $F = 3.85$) in the Lorenz-96 model for $n = 24$. The value of $x_j(t)$ is plotted as a function of $t$ and $j$. For visualization purposes linear interpolation between $x_j$ and $x_{j+1}$ has been applied in order to make the diagram continuous in the variable $j$.




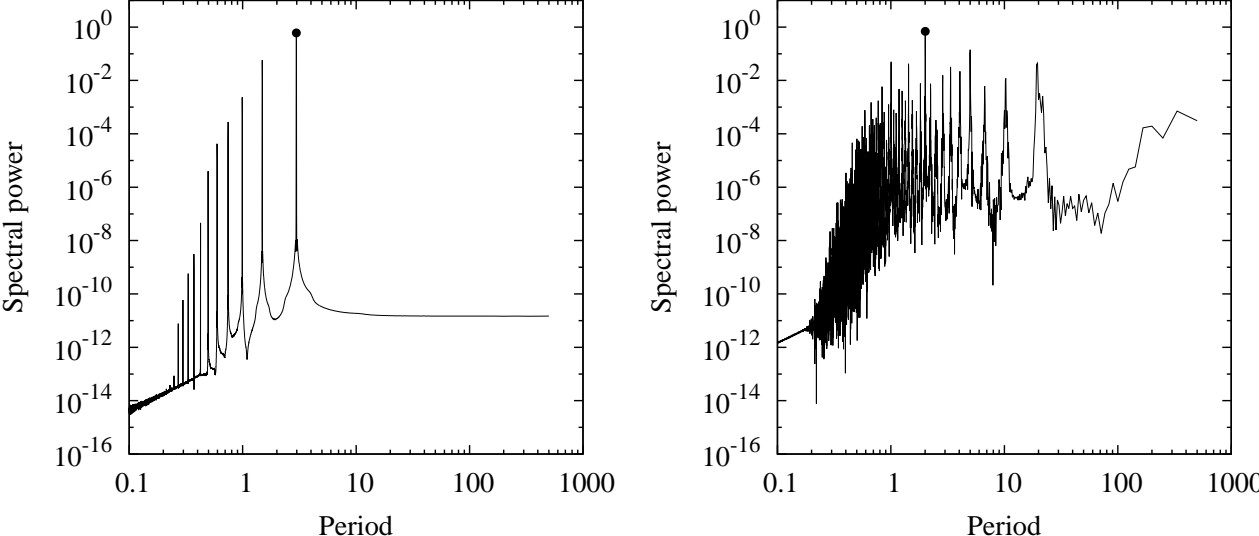

**Figure 2.** Power spectra of the attractors of Fig. 1. Note that the maximum spectral power (indicated by a circle) is attained at nearly the same period.

In this equation, $\mu \in \mathbb{R}$ is a parameter modeling external circumstances such as forcing. Assume that for the parameter value $\mu_0$ the system has an equilibrium solution $\boldsymbol{x}_0$, i.e. $\boldsymbol{f}(\boldsymbol{x}_0, \mu_0) = \boldsymbol{0}$. In the context of geophysics $\boldsymbol{x}_0$ represents a steady flow, and its linear stability is determined by the eigenvalues of the Jacobian matrix $D\boldsymbol{f}(\boldsymbol{x}_0, \mu_0)$. An equilibrium becomes unstable when eigenvalues of the Jacobian matrix cross the imaginary axis upon variation of the parameter $\mu$. Dijkstra (2005) provides

5   an extensive discussion of the physical interpretation of bifurcation behaviour.

Assume that $D\boldsymbol{f}(\boldsymbol{x}_0, \mu_0)$ has eigenvalues $\pm\omega i$. Without loss of generality we may assume that the corresponding complex eigenvectors $\boldsymbol{u} \pm i\boldsymbol{v}$ have unit length. If the equilibrium $\boldsymbol{x}_0$ is stable for $\mu < \mu_0$ and unstable for $\mu > \mu_0$, then under suitable nongenericity conditions a stable periodic orbit exists for $\mu > \mu_0$ (Kuznetsov, 2004). For small values of $\varepsilon = \sqrt{\mu - \mu_0}$ the periodic orbit that is born at the Hopf bifurcation can be approximated by

10   $$\boldsymbol{x}(t) = \boldsymbol{x}_0 + \varepsilon \operatorname{Re}\left[(\boldsymbol{u} + i\boldsymbol{v})e^{i\omega t}\right] + O(\varepsilon^2), \tag{3}$$

see Beyn et al. (2002). In the context of geophysical applications this first-order approximation of the periodic orbit can be interpreted as a wave-like perturbation imposed on a steady mean flow. The spatiotemporal properties of this wave can now be determined by the vectors $\boldsymbol{x}_0, \boldsymbol{u}, \boldsymbol{v}$ and the frequency $\omega$.



| Reference | Application | $n$ |
|---|---|---|
| Basnarkov and Kocarev (2012) | Forecast improvement | 960 |
| Danforth and Yorke (2006) | Forecasting in chaotic systems | 40 |
| Dieci et al. (2011) | Computing Lyapunov exponents | 40 |
| Gallavotti and Lucarini (2014) | Non-equilibrium ensembles | 32 |
| Hallerberg et al. (2010) | Bred vectors | 1024 |
| Hansen and Smith (2000) | Operational constraints | 40 |
| Haven et al. (2005) | Predictability | 40 |
| Karimi and Paul (2010) | Chaos | $4, \ldots, 50$ |
| De Leeuw et al. (2017) | Data assimilation | 36 |
| Lorenz (2006) | Predictability | $4, 36$ |
| Lorenz (2005) | Designing chaotic models | 30 |
| Lorenz and Emanuel (1998) | Data assimilation | 40 |
| Lucarini and Sarno (2011) | Ruelle linear response theory | 40 |
| Orrell et al. (2001) | Model error | 8 |
| Orrell (2002) | Metric in forecast error growth | 8 |
| Orrell and Smith (2003) | Spectral bifurcation diagrams | $4, 8, 40$ |
| Ott et al. (2004) | Data assimilation | $40, 80, 120$ |
| Pazó et al. (2008) | Spatiotemporal chaos | 128 |
| Stappers and Barkmeijer (2012) | Adjoint modelling | 40 |
| Sterk and Van Kekem (2017) | Predictability of extremes | $4, 7, 24$ |
| Sterk et al. (2012) | Predictability of extremes | 36 |
| Trevisan and Palatella (2011) | Data assimilation | $40, 60, 80$ |

**Table 1.** Recent papers with applications of the Lorenz-96 model and the values of $n$ that were used.

## 3 Waves in the Lorenz-96 model

In this section we study waves in the Lorenz-96 model and how their spatiotemporal characteristics depend on the parameters $n$ and $F$.

### 3.1 Traveling waves for $n \geq 4$ and $F > 0$

5   For all $n \in \mathbb{N}$ and $F \in \mathbb{R}$ the point $\boldsymbol{x}_F = (F, \ldots, F)$ is an equilibrium solution of Eq. (1). This equilibrium represents a steady flow, and since all components are equal the flow is spatially uniform. The stability of $\boldsymbol{x}_F$ is determined by the eigenvalues of the Jacobian matrix of Eq. (1). Note that the Lorenz-96 model is invariant under the symmetry $x_i \rightarrow x_{i+1}$ while taking into account the periodic boundary condition. As a consequence the Jacobian matrix evaluated at $\boldsymbol{x}_F$ is circulant which means that each row is a right cyclic shift of the previous row, and so the matrix is completely determined by its first row. If we denote





this row by

$$(c_0, c_1, \ldots, c_{n-1}),$$

then it follows from Gray (2006) that the eigenvalues of the circulant matrix can be expressed in terms of roots of unity $\rho_j = \exp(-2\pi i j/n)$ as follows:

$$\lambda_j = \sum_{k=0}^{n-1} c_k \rho_j^k, \qquad j = 0, \ldots, n-1. \tag{4}$$

An eigenvector corresponding to $\lambda_j$ is given by

$$\boldsymbol{v}_j = \frac{1}{\sqrt{n}} \begin{pmatrix} 1 & \rho_j & \rho_j^2 & \cdots & \rho_j^{n-1} \end{pmatrix}^\top.$$

In particular, for the Lorenz–96 model Eq. (1) we have that the Jacobian matrix at $\boldsymbol{x}_F$ has only three nonzero elements on its first row, viz. $c_0 = -1$, $c_1 = F$, $c_{n-2} = -F$. Hence, the eigenvalues $\lambda_j$ can be expressed in terms of $n$ and $F$ as follows:

$$\lambda_j = -1 + Ff(2\pi j/n) + Fg(2\pi j/n)i, $$

where the functions $f$ and $g$ are defined as

$$f(x) = \cos(x) - \cos(2x),$$
$$g(x) = -\sin(x) - \sin(2x). \tag{5}$$

For $F = 0$ the equilibrium $\boldsymbol{x}_F$ is stable as $\operatorname{Re}\lambda_j = -1$ for all $j = 0, \ldots, n-1$. The real part of the eigenvalue $\lambda_j$ changes sign if the equation

$$F = \frac{1}{f(2\pi j/n)} \tag{6}$$

is satisfied. The graph of $f$ in Fig. 3 shows that for $F > 0$ Eq. (6) can have at most four solutions. Since $f$ is symmetric around $x = \pi$ it follows that if $j$ is a solution of Eq. (6) then so is $n - j$. This means that the equilibrium $\boldsymbol{x}_F$ becomes unstable for $F > 0$ when either a pair or a double pair of eigenvalues becomes purely imaginary. The main result is summarized in the following theorem.

**Theorem 1.** *Assume that $n \geq 4$ and $l \in \mathbb{N}$ satisfies $0 < l < \frac{n}{2}, l \neq \frac{n}{3}$. Then the $l$-th eigenvalue pair $(\lambda_l, \lambda_{n-l})$ of the trivial equilibrium $\boldsymbol{x}_F$ crosses the imaginary axis at the parameter value $F_H(l, n) := 1/f(2\pi l/n)$ and thus $\boldsymbol{x}_F$ bifurcates through either a Hopf or a double-Hopf bifurcation. A double-Hopf bifurcation, with two pairs of eigenvalues crossing the imaginary axis, occurs if and only if there exist $l_1, l_2 \in \mathbb{N}$ such that*

$$\cos\left(\frac{2\pi l_1}{n}\right) + \cos\left(\frac{2\pi l_2}{n}\right) = \frac{1}{2}. \tag{7}$$

*Otherwise, a Hopf bifurcation occurs. Moreover, the first Hopf bifurcation of $\boldsymbol{x}_F$ is always supercritical.*





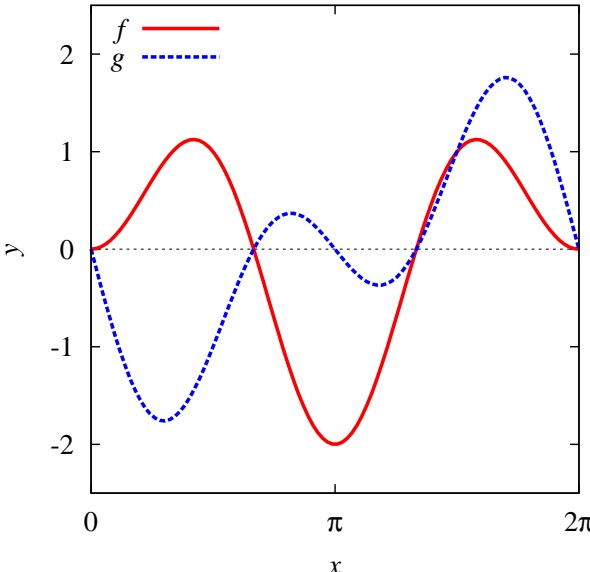

**Figure 3.** Graphs of the functions $f$ and $g$ as defined in Eq. (5). The eigenvalues of the equilibrium $\boldsymbol{x}_F = (F, \dots, F)$ are given by $\lambda_j = -1 + Ff(2\pi j/n) + Fg(2\pi j/n)i$ for $j = 0, \dots, n-1$. The shapes of the graphs of $f$ and $g$ imply that the equilibrium $\boldsymbol{x}_F$ can only lose stability through either a Hopf or a double-Hopf bifurcation for $F > 0$.

The proof of Theorem 1 can be found in Van Kekem and Sterk (2017a) in which also an expression for the first Lyapunov coefficient is derived which determines for which $l$ the Hopf bifurcation is sub- or supercritical. Observe that Theorem 1 implies that a double-Hopf bifurcation occurs for $n = 10m$ (with $l_1 = m, l_2 = 3m, F = 2$) and $n = 12m$ (with $l_1 = 2m$, $l_2 = 3m$, $F = 1$). In Sect. 4 we will explain how double-Hopf bifurcations lead to the coexistence of two or more stable traveling waves

5    with different wave numbers.

From the eigenvalues that cross the imaginary axis and the corresponding eigenvectors we can deduce the physical characteristics of the periodic orbit that arises after a Hopf bifurcation. When the $l$-th eigenvalue pair $(\lambda_l, \lambda_{n-l})$ crosses the imaginary axis we can write

$$\lambda_l = \frac{g(2\pi l/n)}{f(2\pi l/n)}i = -\cot(\pi l/n)i, \quad \lambda_{n-l} = \bar{\lambda}_l = \cot(\pi l/n)i.$$

10    If we set $\omega = \cot(\pi l/n)$, then according to Eq. (3) an approximation of the periodic orbit is given by

$$\begin{aligned}
x_j(t) &= F + \varepsilon \operatorname{Re} \frac{e^{i(\omega t - 2\pi ijl/n)}}{\sqrt{n}} + O(\varepsilon^2) \\
&= F + \frac{\varepsilon}{\sqrt{n}} \cos(\omega t - 2\pi jl/n) + O(\varepsilon^2).
\end{aligned}$$





This is indeed the expression for a *traveling* wave in which the spatial wave number and the period are given by respectively $l$ and $T = 2\pi/\omega = 2\pi\tan(\pi l/n)$. Thus the index of the eigenpair that crosses the imaginary axis determines the propagation characteristics of the wave.

Note that Hopf bifurcations of an *unstable* equilibrium will result in an unstable periodic orbit. Therefore, not all waves that
are guaranteed to exist by Theorem 1 will be visible in numerical experiments. Equation (6) implies that for $F > 0$ the *first* Hopf bifurcation occurs for the eigenpair $(\lambda_l, \lambda_{n-l})$ with index

$$l_1^+(n) = \underset{0<j<n/3}{\arg\max} f(2\pi j/n). \tag{8}$$

In Appendix A1 it is shown that, except for $n = 7$, the integer $l_1^+(n)$ satisfies the bounds

$$\frac{n}{6} \le l_1^+(n) \le \frac{n}{4}, \tag{9}$$

which means that the wave number increases linearly with the dimension $n$. Since the function $f$ has a maximum at $x = \arccos(\frac{1}{4})$ we have

$$\lim_{n\to\infty} \frac{2\pi l_1^+(n)}{n} = \arccos(\tfrac{1}{4}),$$

which is consistent with Eq. (9). As a corollary we find that the period of this wave tends to a finite limit as $n \to \infty$:

$$T_\infty = \lim_{n\to\infty} 2\pi\tan\left(\frac{\pi l_1^+(n)}{n}\right) = 2\pi\tan(\tfrac{1}{2}\arccos(\tfrac{1}{4})) \approx 4.867.$$

Figure 4 shows a graph of the period and the wave number as a function of $n$. Note that the period settles down on the value $T_\infty$.

### 3.2   Traveling waves for odd $n \ge 4$ and $F < 0$

Now assume that $n$ is odd. For $F < 0$ Eq. (6) has precisely two solutions which implies that the first bifurcation of $\boldsymbol{x}_F$ is a supercritical Hopf bifurcation. The index of the first bifurcating eigenpair $(\lambda_l, \lambda_{n-l})$ follows by minimizing the value of the
function $f$ in Eq. 5:

$$l_1^-(n) = \frac{n-1}{2}.$$

Again, the wave number increases linearly with $n$, but at a faster rate than in the case $F > 0$. Now the period of the wave is given by

$$T = 2\pi\tan\left(\frac{\pi(n-1)}{2n}\right) = O(4n),$$

where the last equality sign follows from the computations in Appendix A2. This implies that contrary to the case $F > 0$ the period increases monotonically with $n$ and does not tend to a limiting value as $n \to \infty$.

Note that for even $n$ and $F < 0$ the first bifurcation is *not* a Hopf bifurcation since $\lambda_{n/2} = -1 - 2F$ is a *real* eigenvalue that changes sign at $F = \frac{1}{2}$. Surprisingly, the case $n = 4$ is not analytically tractable. The case $n = 6$ will be studied analytically in Sect. 3.3. In Sect. 3.4 we will numerically study the bifurcations for other values of $n$ and $F < 0$.



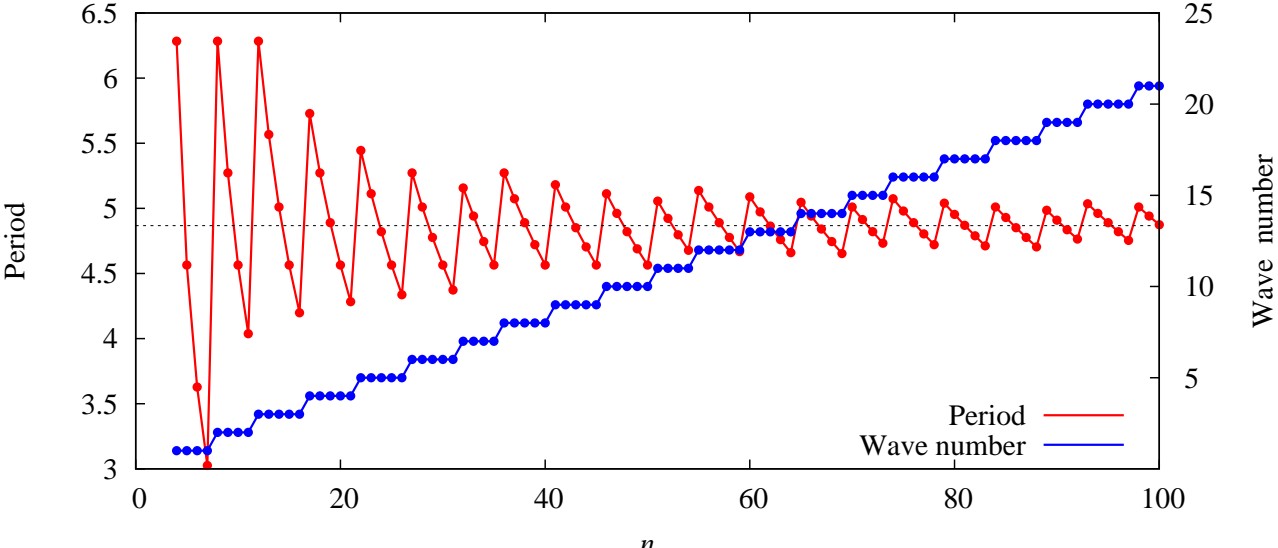

**Figure 4.** As the equilibrium $\boldsymbol{x}_F = (F, \ldots, F)$ loses stability through a (double-)Hopf bifurcation for $F > 0$ a periodic attractor is born which represents a traveling wave. The spatial wave number increases linearly with $n$, whereas the period tends to a finite limit.

### 3.3 Standing waves for $n = 6$ and $F < 0$

We now consider the dimension $n = 6$. At $F = -\frac{1}{2}$ the eigenvalue $\lambda_3$ changes sign. Note that the equilibrium $\boldsymbol{x}_F$ *cannot* exhibit a saddle-node bifurcation since $\boldsymbol{x}_F$ continues to exist for $F < -\frac{1}{2}$. Instead, at $F = -\frac{1}{2}$ there must be a branching point which is associated to either a pitchfork or a transcritical bifurcation. If we try for $F < -\frac{1}{2}$ an equilibrium solution of the form $x_P = (a, b, a, b, a, b)$ then it follows that $a$ and $b$ must satisfy the equations

$$b(b - a) - a + F = 0,$$
$$a(a - b) - b + F = 0.$$

Of course $a = b = F$ is a solution to these equations, but this would lead to the already known equilibrium $\boldsymbol{x}_F = (F, F, F, F, F, F)$. There is an additional pair of solutions which is given by

$$a = \frac{-1 + \sqrt{-1 - 2F}}{2},$$
$$b = \frac{-1 - \sqrt{-1 - 2F}}{2}. \tag{10}$$

With these values of $a$ and $b$ we obtain two new equilibria $x_{P,1} = (a, b, a, b, a, b)$ and $x_{P,2} = (b, a, b, a, b, a)$ that exist for $F < -\frac{1}{2}$ in addition to the equilibrium $\boldsymbol{x}_F$. This means that a pitchfork bifurcation occurs at $F = -\frac{1}{2}$.



As $F$ decreases, each of the new equilibria $x_P^{1,2}$ may bifurcate again. We first consider the equilibrium $x_{P,1}$ for which the Jacobian matrix is given by

$$
J = \begin{pmatrix}
-1 & b & 0 & 0 & -b & b-a \\
a-b & -1 & a & 0 & 0 & -a \\
-b & b-a & -1 & b & 0 & 0 \\
0 & -a & a-b & -1 & a & 0 \\
& 0 & -b & b-a & -1 & b \\
a & 0 & 0 & -a & a-b & -1
\end{pmatrix}.
$$

Note that $J$ is no longer circulant: in addition to shifting each row in a cyclic manner, the values of $a$ and $b$ also need to be interchanged. In particular, this means that the eigenvalues can no longer be determined by means of Eq. (4). Symbolic manipulations with the computer algebra package Mathematica (Wolfram Research, Inc., 2016) show that an eigenvalue crossing occurs for $F = -\frac{7}{2}$ in which case $a = \frac{1}{2}(-1+\sqrt{6})$ and $b = \frac{1}{2}(-1-\sqrt{6})$ and the characteristic polynomial of $J$ is given by

$$
\det(J - \lambda I) = 468 + 219\lambda + 246\lambda^2 + 91\lambda^3 + 33\lambda^4 + 6\lambda^5 + \lambda^6
$$
$$
= (3 + \lambda^2)(12 + \lambda + \lambda^2)(13 + 5\lambda + \lambda^2).
$$

This expression shows that $J$ has two purely imaginary eigenvalues $\pm i\sqrt{3}$ and the remaining four complex eigenvalues have negative real part. Therefore the equilbrium $x_{P,1}$ undergoes a Hopf bifurcation at $F = -\frac{7}{2}$. Numerical experiments with Mathematica show that the matrix $J - i\sqrt{3}I$ has a null vector of the form

$$
v = \begin{pmatrix} v_0 & v_1 & v_0 e^{2\pi i/3} & v_1 e^{2\pi i/3} & v_0 e^{-2\pi i/3} & v_1 e^{-2\pi i/3} \end{pmatrix}^\top,
$$

where we can take

$v_0 = 6\sqrt{2} + 2i$   and   $v_1 = 3\sqrt{2} + 5\sqrt{3} - (5 + \sqrt{6})i.$

Hence, using Eq. (3) the periodic orbit can be approximated as

$$
x_0(t) = \frac{-1+\sqrt{6}}{2} + \frac{\varepsilon}{\|v\|} \operatorname{Re} v_0 e^{i\sqrt{3}t} + O(\varepsilon^2),
$$
$$
x_1(t) = \frac{-1-\sqrt{6}}{2} + \frac{\varepsilon}{\|v\|} \operatorname{Re} v_1 e^{i\sqrt{3}t} + O(\varepsilon^2),
$$
$$
x_2(t) = \frac{-1+\sqrt{6}}{2} + \frac{\varepsilon}{\|v\|} \operatorname{Re} v_0 e^{i(\sqrt{3}t+2\pi/3)} + O(\varepsilon^2),
$$
$$
x_3(t) = \frac{-1-\sqrt{6}}{2} + \frac{\varepsilon}{\|v\|} \operatorname{Re} v_1 e^{i(\sqrt{3}t+2\pi/3)} + O(\varepsilon^2),
$$
$$
x_4(t) = \frac{-1+\sqrt{6}}{2} + \frac{\varepsilon}{\|v\|} \operatorname{Re} v_0 e^{i(\sqrt{3}t-2\pi/3)} + O(\varepsilon^2),
$$
$$
x_5(t) = \frac{-1-\sqrt{6}}{2} + \frac{\varepsilon}{\|v\|} \operatorname{Re} v_1 e^{i(\sqrt{3}t-2\pi/3)} + O(\varepsilon^2).
$$

(11)



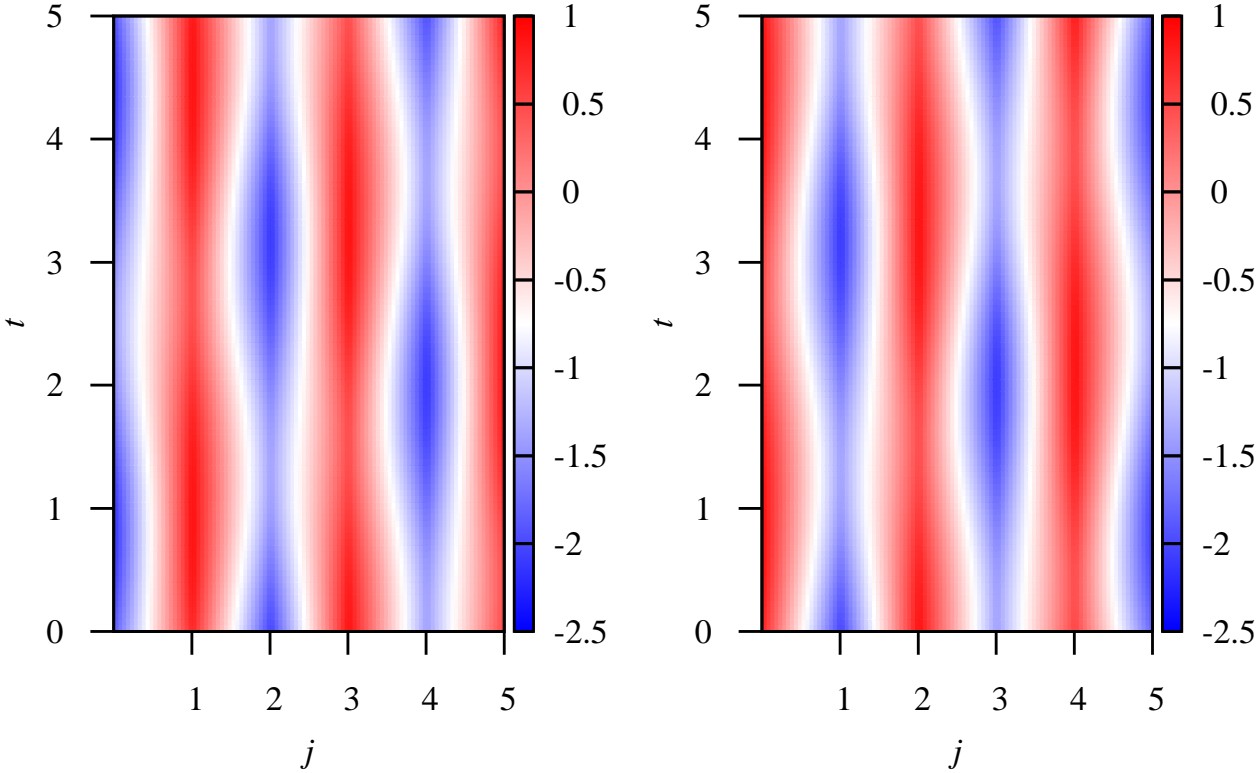

**Figure 5.** As Fig. 1, but for two periodic attractors for $n = 6$ and $F = -3.6$. These attractors are born at Hopf bifurcations of the equilibria $x_{P,1}$ (left) and $x_{P,2}$ (right) at $F = -7/2$. Note that the waves do not travel "eastward" or "westward". The pitchfork bifurcation changed the mean flow which in turn changes the propagation of the wave.

Note that if $\varepsilon = \sqrt{-\frac{7}{2} - F}$ is sufficiently small, then $x_j(t)$ is always positive (resp. negative) for $j = 0, 2, 4$ (resp. $j = 1, 3, 5$). This implies that the periodic orbit represents a *standing* wave rather than a traveling wave. The period of the wave is $T = 2\pi/\sqrt{3}$ and the spatial wave number is 3. These spatiotemporal properties are clearly visible in the left panel of Fig. 5.

5      The computations for the equilibrium $x_{P,2}$ are similar and show that *another* Hopf bifurcation takes place at $F = -\frac{7}{2}$. This means that for $F < -\frac{7}{2}$ there exists a *second* stable period orbit which coexists with the stable periodic orbit born at the Hopf bifurcation of $x_{P,1}$. Its first-order approximation is almost identical to Eq. (11): only the numerators $1 - \sqrt{6}$ and $1 + \sqrt{6}$ need to be interchanged and therefore the complete expression will be omitted. Hence, the two coexisting stable waves that arise from the two Hopf bifurcations of the equilibria $x_{P,1}$ and $x_{P,2}$ have the same spatiotemporal properties, but they differ in spatial



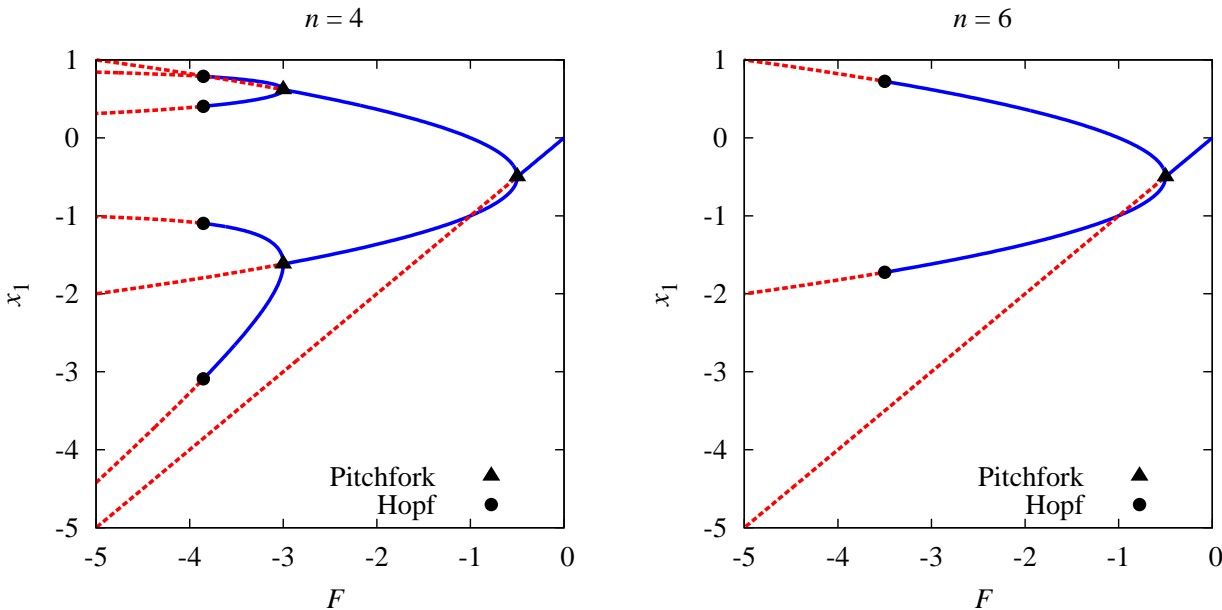

**Figure 6.** Bifurcation diagrams obtained by continuation of the equilibrium $\boldsymbol{x}_F = (F, \dots, F)$ for $F < 0$ for the dimensions $n = 4$ and $n = 6$. Stable (unstable) branches are marked by solid (dashed) lines. For $n = 4$ two pitchforks in a row occur before the Hopf bifurcation, whereas for $n = 6$ only one pitchfork occurs before the Hopf bifurcation. The bifurcation diagram for $n = 4k$ with $k \in \mathbb{N}$ (resp. $n = 4k + 2$) is qualitatively similar to the bifurcation diagram for $n = 4$ (resp. $n = 6$), see the main text.

phase which is indeed visible in the Hovmöller diagrams in Fig. 5. These results show how the pitchfork bifurcation changed the mean flow and hence also the propagation characteristics of the wave. In the next section we will explore spatiotemporal properties of waves for $F < 0$ and other even values of $n$.

5 **3.4 Standing waves for even $n \geq 4$ and $F < 0$**

The case $n = 4$ turns out to be more complicated than the case $n = 6$. If $n = 4$, then the equilibrium $\boldsymbol{x}_F = (F, F, F, F)$ undergoes a pitchfork bifurcation at $F = -\frac{1}{2}$ since $\lambda_2 = 0$. Just as in the case $n = 6$ two new branches of equilibria appear which are given by

$$x_{P,1} = (a, b, a, b), \quad x_{P,2} = (b, a, b, a),$$

10 where $a, b$ are again given by Eq. (10). The Jacobian matrix at the equilibrium $x_{P,1}$ is given by

$$J = \begin{pmatrix} -1 & b & -b & b-a \\ a-b & -1 & a & -a \\ -b & b-a & -1 & b \\ a & -a & a-b & -1 \end{pmatrix}$$




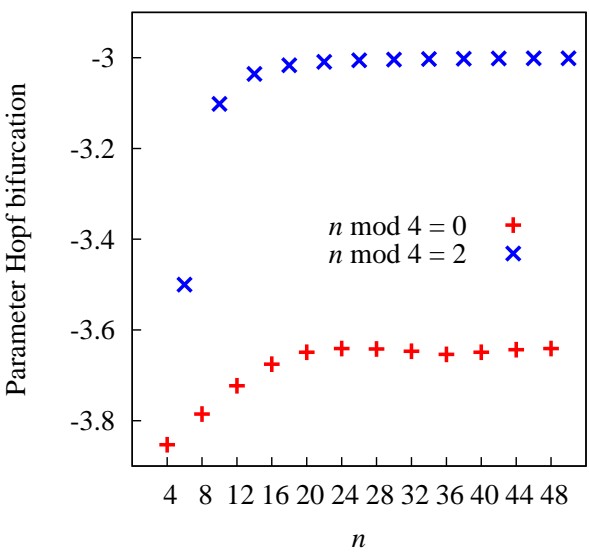
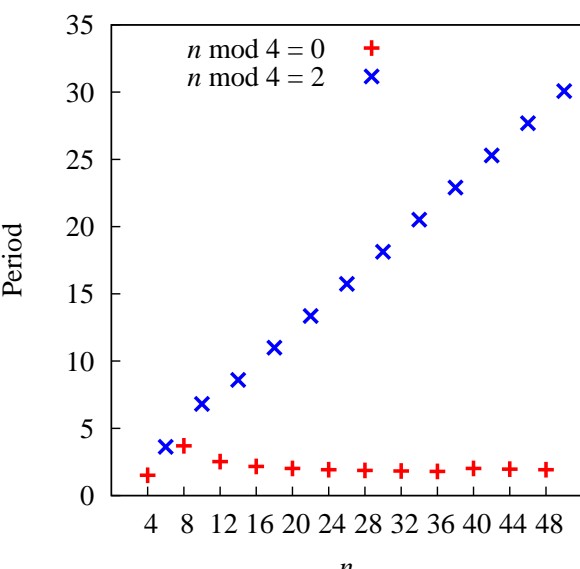

**Figure 7.** Parameter values of the first Hopf bifurcation (left) and the periods of the periodic attractor (right) that appears after the Hopf bifurcation for $F < 0$ and even values of the dimension $n$. For clarity the cases $n = 4k$ and $n = 4k + 2$ have been marked with different symbols in order to emphasize the differences.

For $F = -3$, in which case $a = (-1 + \sqrt{5})/2$ and $b = (-1 - \sqrt{5})/2$, the characteristic polynomial of the matrix $J$ is given by

$$\det(J - \lambda I) = \lambda(30 + 13\lambda + 4\lambda^2 + \lambda^3),$$

which implies that a real eigenvalue of $J$ becomes zero at $F = -3$. For the equilibrium $x_{P,2}$ we obtain the same result. Since the equilibria $x_{P,1}$ and $x_{P,2}$ continue to exist for $F < -3$ a saddle-node bifurcation is ruled out. Numerical continuation using the software package AUTO-07p (Doedel and Oldeman, 2007) shows that again a pitchfork bifurcation takes place at $F = -3$. It is not feasible to derive analytic expressions for the new branches of equilibria as in Eq. (10). Continuation of the four branches while monitoring their stability indicates that at $F \approx -3.853$ in total four Hopf bifurcations occur (one at each branch). Figure 6 shows the bifurcation diagrams for the cases $n = 4$ and $n = 6$.

The question is whether the results described above persist for even dimensions $n > 6$. To that end we conducted the following numerical experiment. For all even dimensions $4 \leq n \leq 50$ we used the software package AUTO-07p to numerically continue the equilibrium $\boldsymbol{x}_F = (F, \ldots, F)$ for $F < 0$ while monitoring the eigenvalues to detect bifurcations. At each pitchfork bifurcation we performed a branch switch in order to follow the new branches of equilibria and detect their bifurcations. Once a Hopf bifurcation is detected we can compute the period of the wave as $T = 2\pi/\omega$ from the eigenvalue pair $\pm\omega i$. The results of this experiment reveal that the cases $n = 4k$ and $n = 4k + 2$ are different both qualitatively and quantitatively.





If $n = 4k + 2$ for some $k \in \mathbb{N}$, then one pitchfork bifurcation occurs at $F = -0.5$. This follows directly from Eq. (4) for the eigenvalues of the equilibrium $\boldsymbol{x}_F$: for even $n$ we have $\lambda_{n/2} = -1 - 2F$ which changes sign at $F = -\frac{1}{2}$. From the pitchfork bifurcation two new branches of stable equilibria emanate. Each of these equilibria is of the form

$$(a, b, a, b, a, b, \dots) \tag{12}$$

with $a > 0$ and $b < 0$; the other equilibrium just follows by interchanging $a$ and $b$. Each of the two equilibria undergoes a Hopf bifurcation, which leads to the coexistence of two stable waves. Figure 7 (left panel) suggests that the value of $F$ at which this bifurcation occurs is not constant, but tends to $-3$ as $n \to \infty$. The period of the periodic attractor that is born at the Hopf bifurcation increases almost linearly with $n$: fitting the function $T(n) = \alpha + \beta n$ to the numerically computed periods gives $\alpha = 0.36$ and $\beta = 0.59$, see Fig. 7 (right panel).

If $n = 4k$ for some $k \in \mathbb{N}$, then *two* Pitchfork bifurcations in a row occur at $F = -0.5$ and $F = -3$. After the second pitchfork bifurcation there are four branches of equilibria. Each of these equilibria is of the form

$$(a, b, c, d, a, b, c, d, \dots) \tag{13}$$

where $a, b, c, d$ alternate in sign; the other equilibria are obtained by applying a circulant shift. Each of the four stable equilibria undergoes a Hopf bifurcation at the same value of the parameter $F$, which leads to the coexistence of four stable waves. Figure 7 (left panel) suggests that the value of $F$ at which this bifurcation occurs is not constant, but tends to $-3.64$ as $n \to \infty$. Contrary to the case $n = 4k + 2$, the period of the periodic attractor that appears after the Hopf bifurcation settles down and tends to $1.92$ as $n \to \infty$.

In spite of the aforementioned quantitative differences between the cases $n = 4k + 2$ and $n = 4k$, the wave numbers depend in the same way on $n$ in both cases. Equations (12) and (13) show that the $n$ components of the equilibrium that undergoes the Hopf bifurcation alternate in sign. Therefore, sufficiently close to the Hopf bifurcation the components $x_0(t), \dots, x_{n-1}(t)$ of the periodic orbit will also alternate in sign. Hence, the resulting standing waves consists of $n/2$ "troughs" and "ridges" which means that their wave number equals $n/2$.

## 4 Multi-stability: coexistence of waves

The results of Sect. 3.4 show that for even $n$ and $F < 0$ either two or four stable periodic orbits coexist for the same parameter values. This phenomenon is referred to as *multi-stability* in the dynamical systems literature. An overview of the wide range of applications of multi-stability in different disciplines of science is given by Feudel (2008).

Multi-stability also occurs for $F > 0$ but due to a very different reason. For $n = 12$ Theorem 1 implies that the first bifurcation of the equilibrium $\boldsymbol{x}_F = (F, \dots, F)$ for $F > 0$ is *not* a Hopf bifurcation, but a double-Hopf bifurcation. Indeed, at $F = 1$ we have two pairs of purely imaginary eigenvalues, namely $(\lambda_2, \lambda_{10}) = (-i\sqrt{3}, i\sqrt{3})$ and $(\lambda_3, \lambda_9) = (-i, i)$. Note that the double-Hopf bifurcation is a codimension-2 bifurcation which means that generically two parameters must be varied in order for the bifurcation to occur (Kuznetsov, 2004). However, symmetries such as those in the Lorenz-96 model can reduce the codimension of a bifurcation.





In previous work (Van Kekem and Sterk, 2017a) we have introduced an embedding of the Lorenz-96 model in a 2-parameter family by adding a diffusion-like term multiplied by an additional parameter $G$:

$$\frac{dx_j}{dt} = x_{j-1}(x_{j+1} - x_{j-2}) - x_j + G(x_{j-1} - 2x_j + x_{j+1}) + F, \quad j = 0, \ldots, n-1. \tag{14}$$

Note that by setting $G = 0$ we retrieve the original Lorenz-96 model in Eq. (1). Since the Jacobian matrix of Eq. (14) is again a circulant matrix we can use Eq. (4) to determine its eigenvalues:

$$\lambda_j = -1 - 2G(1 - \cos(2\pi j/n)) + Ff(2\pi j/n) + Fg(2\pi j/n)i \tag{15}$$

Also note that $\boldsymbol{x}_F = (F, \ldots, F)$ remains an equilibrium solution of Eq. (14) for all $(F, G)$. The Hopf bifurcations of $\boldsymbol{x}_F$ described in Theorem 1 now occur along the lines

$$G = \frac{Ff(2\pi j/n) - 1}{2(1 - \cos(2\pi j/n))}, \tag{16}$$

and the intersection of two such lines leads to a double-Hopf bifurcation.

Figure 8 shows a local bifurcation diagram of the 2-parameter Lorenz-96 model in the $(F, G)$-plane for $n = 12$ which was numerically computed using MATCONT (Dhooge et al., 2011). A double-Hopf point is located at $(F, G) = (1, 0)$ which is indeed implied by Theorem 1. Computing the normal form of this bifurcation shows that the unfolding of the bifurcation is of "type I" as described by Kuznetsov (2004). This means that from the double-Hopf point only two curves of Neĭmark-Sacker bifurcations emanate. These curves bound a region of the $(F, G)$-plane in which two stable periodic attractors coexist with an unstable 2-torus. We will refer to this region as the "multi-stability lobe". Figure 9 shows two periodic attractors with wave numbers 2 and 3 in the multi-stability lobe for $n = 12$ and $(F, G) = (1.5, 0)$.

Double-Hopf bifurcations are abundant in the 2-parameter Lorenz-96 model of Eq. (14). The lines described in Eq. (16) have a different slope for all $0 < j < n/2$ and $j \neq n/3$, and hence they mutually intersect each other. This implies that the number of double-Hopf points in the $(F, G)$-plane grows quadratically with $n$, see Appendix A3. However, not all these points will have an influence on the dynamics: if $\boldsymbol{x}_F$ is already unstable, then any dynamical object born through the double-Hopf bifurcation will also be unstable. In what follows, we only consider the double-Hopf bifurcations through which $\boldsymbol{x}_F$ can change from stable to unstable. We can find such points as follows. Starting from the line in Eq. (16) with $j = l_1^+(n)$ as defined by Eq. (8), we first compute double-Hopf points by computing the intersections with all other lines. From these intersections we select those that satisfy the condition $\max\{\text{Re}\,\lambda_j : j = 0, \ldots, n-1\} = 0$.

Figure 10 shows the $G$-coordinates of these double-Hopf points as a function of $n$. Clearly, for large $n$ there exist double-Hopf points which are very close to the $F$-axis which suggests that the multi-stability lobe that emanates from such points can intersect the $F$-axis and hence influence the dynamics of the original Lorenz-96 model for $G = 0$. Moreover, Fig. 10 shows that for $n > 12$ there are always *two* double-Hopf points by which $\boldsymbol{x}_F$ can change from stable to unstable. It is then possible that two multi-stability lobes intersect each other which leads to a region in the $(F, G)$-plane in which at least three stable waves coexist.

Figures 11–13 show bifurcation diagrams of three periodic orbits as a function of $F$ for $G = 0$ for $n = 40, 60, 80$. For each periodic orbit the continuation is started from a Hopf bifurcation of the equilibrium $\boldsymbol{x}_F$. If $\boldsymbol{x}_F$ is unstable, then so will be the

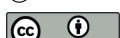



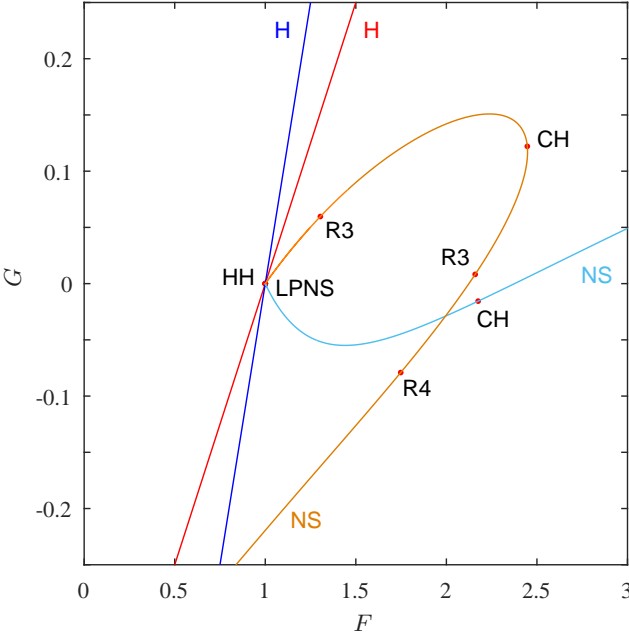

**Figure 8.** Bifurcation diagram of the 2-parameter system (14) in the $(F, G)$-plane for $n = 12$. A double-Hopf bifurcation point is located at the point $(F, G) = (1, 0)$ due to the intersection of two Hopf bifurcation lines. From this codimension-2 point two Neĭmark-Sacker bifurcation curves emanate which bound a "lobe-shaped" region in which two periodic attractors coexist.

periodic orbit. However, when the boundary of a multi-stability lobe is crossed a Neĭmark-Sacker bifurcation occurs by which a periodic orbit can gain stability. For specific intervals of the parameter $F$ three stable periodic orbits coexist. Since Fig. 10 shows that for large values of $n$ the double-Hopf bifurcations are close to the $F$-axis, we expect that the coexistence of three

5    or more stable waves is typical for the Lorenz-96 model.

## 5    Conclusions

In this paper we have studied spatiotemporal properties of waves in the Lorenz-96 model and their dependence on the dimension $n$. For $F > 0$ the first bifurcation of the equilibrium $\boldsymbol{x}_F = (F, \ldots, F)$ is either a supercritical Hopf or a double-Hopf bifurcation and the periodic attractor born at the Hopf bifurcation represents a traveling wave. The spatial wave number is determined by

10   the index of the eigenpair that crosses the imaginary axis and increases linearly with $n$, but the period tends to a finite limit as $n \to \infty$. For $F < 0$ and $n$ *odd* the first bifurcation of $\boldsymbol{x}_F$ is always a supercritical Hopf bifurcation and the periodic attractor that appears after the bifurcation is again a traveling wave. In this case the wave number equals $(n-1)/2$ and the period is $O(4n)$.

For $n$ *even* and $F < 0$ the first bifurcation of $\boldsymbol{x}_F$ is a pitchfork bifurcation which occurs at $F = -\frac{1}{2}$ and leads to two stable equilibria. If $n = 4k + 2$ for some $k \in \mathbb{N}$, then each of these equilibria undergoes a Hopf bifurcation which leads to the



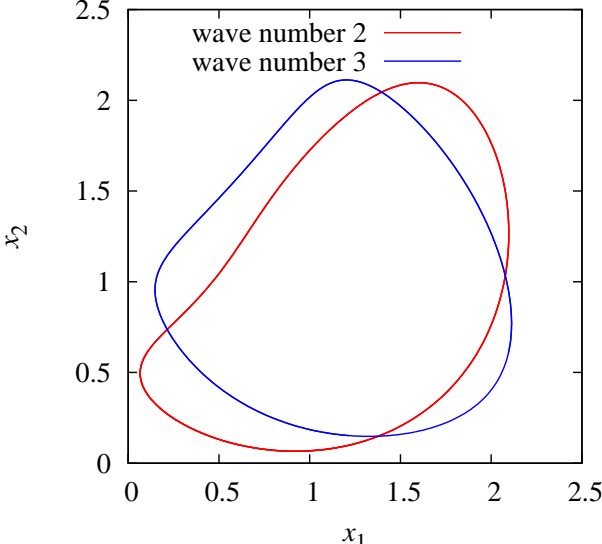

**Figure 9.** Projections onto the $(x_1, x_2)$-plane of coexisting periodic attractors for dimensions $n = 12$ and $(F, G) = (1.5, 0)$, which is inside the multi-stability lobe of Fig. 8.

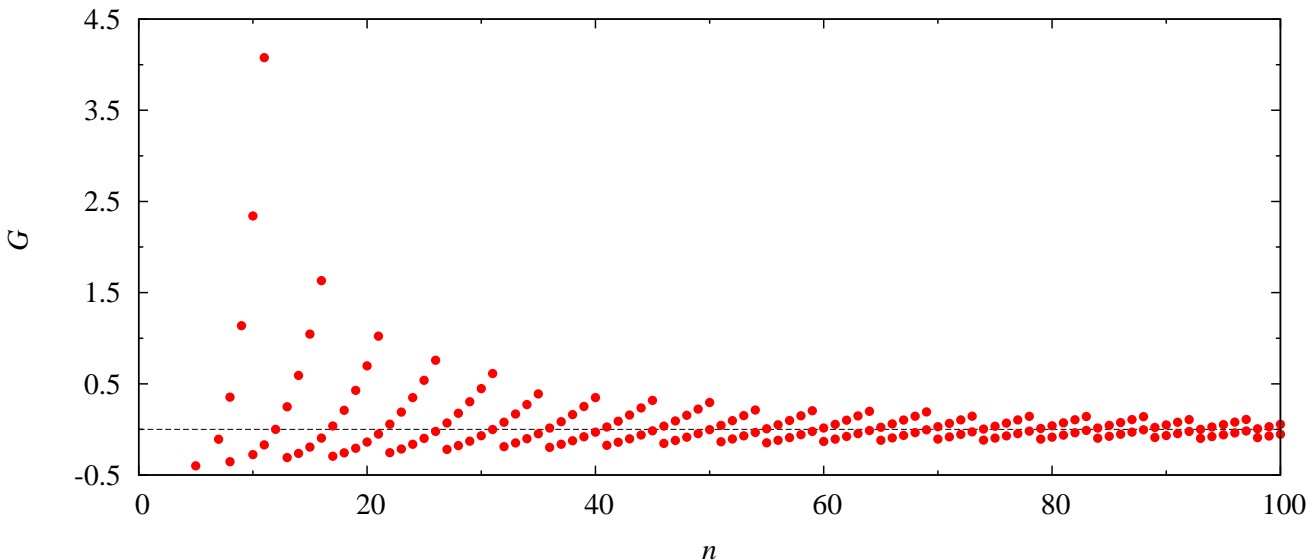

**Figure 10.** $G$-coordinates of double-Hopf points as a function of $n$. Only those double-Hopf points are shown which destabilize the equilibrium $\boldsymbol{x}_F$. For large values of $n$ the double-Hopf bifurcations are close to the $F$-axis in the $(F, G)$-plane, which means that these points are likely to affect the dynamics of the Lorenz-96 model for $G = 0$.


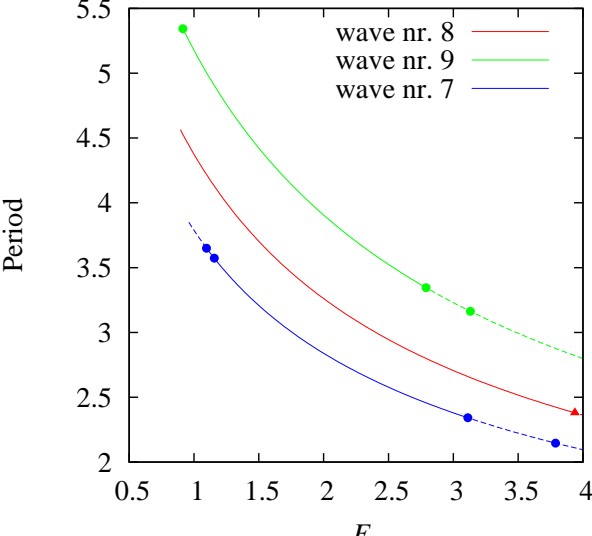

**Figure 11.** Continuation of periodic orbits for $n = 40$ and $G = 0$. The period of the orbit is plotted as a function of $F$. Stable (resp. unstable) orbits are indicated by solid (resp. dashed) lines. Circles denote Neĭmark-Sacker bifurcations and triangles denote period doubling bifurcations. The Hopf bifurcations generating the waves with wave numbers 8, 9, 7 occur at respectively $F = 0.894$, $F = 0.902$, and $F = 0.959$. Clearly, for $1.15 < F < 2.79$ three stable periodic orbits coexist.

coexistence of two standing waves. The role of the pitchfork bifurcation is to change the mean flow which in turn changes the propagation of the wave. If $n = 4k$ for some $k \in \mathbb{N}$, then *two* pitchfork bifurcations take place at $F = -\frac{1}{2}$ and $F = -3$ before a Hopf bifurcation occurs which leads to the coexistence of four standing waves.

The occurrence of pitchfork bifurcations before the Hopf bifurcation leads to multi-stability, i.e. the coexistence of different waves for the same parameter settings. A second scenario that leads to multi-stability is via the double-Hopf bifurcation. For $n = 12$ the equilibrium $\boldsymbol{x}_F$ loses stability through a double-Hopf bifurcation. By adding a second parameter $G$ to the Lorenz-96 model we have studied the unfolding of this codimension-2 bifurcation. Two Neĭmark-Sacker bifurcation curves emanating from the double-Hopf point bound a lobe-shaped region in the $(F, G)$-plane in which two stable traveling waves with different

wave numbers coexist. For dimensions $n > 12$ we find double-Hopf bifurcations *near* the $F$-axis, which can create two multi-stability lobes intersecting each other, and in turn this can lead to the coexistence of *three* stable waves coexisting for $G = 0$ and a range of $F$-values. Hence, adding a parameter $G$ to the Lorenz-96 model helps to explain the dynamics which is observed in the original model for $G = 0$.

Our results provide an overview of the spatiotemporal properties of the Lorenz-96 model for $n \geq 4$ and $F \in \mathbb{R}$. Since the

Lorenz-96 model is often used as a model for testing purposes, our results can be used to select the most appropriate values of $n$ and $F$ for a particular application. The periodic attractors representing traveling or standing waves can bifurcate into chaotic

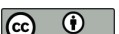


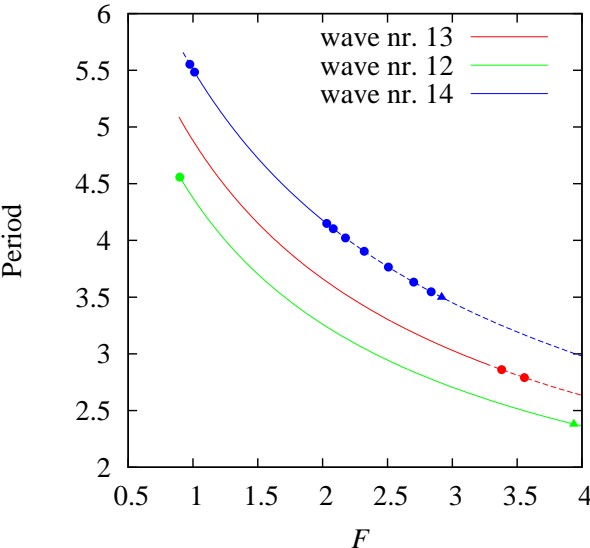

**Figure 12.** As Figure 11, but for $n = 60$. For $1.01 < F < 2.03$ three stable periodic orbits coexist. The Hopf bifurcations generating the waves with wave numbers 13, 12, 14 occur at respectively $F = 0.891$, $F = 0.894$, and $F = 0.923$.

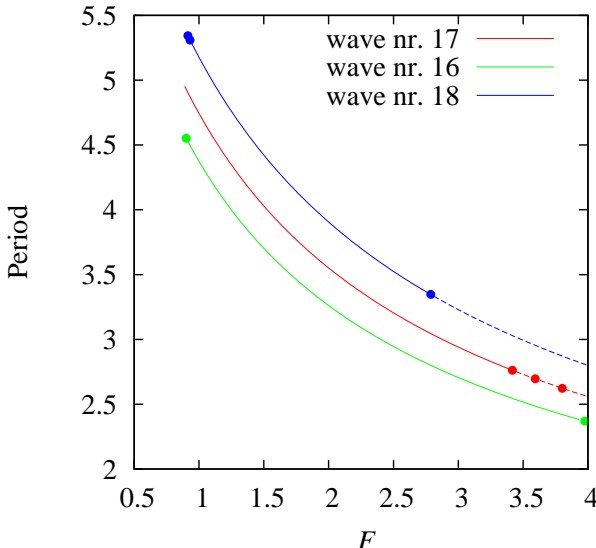

**Figure 13.** As Figure 11, but for $n = 80$. For $0.93 < F < 2.78$ three stable periodic orbits coexist. The Hopf bifurcations generating the waves with wave numbers 17, 16, 18 occur at respectively $F = 0.889$, $F = 0.894$, and $F = 0.902$.





attractors representing irregular versions of these waves, and their spatiotemporal properties are inherited from the periodic attractor, see for example Fig. 1 and Fig. 2. This means that our results on the spatiotemporal properties of waves apply to broader parameter ranges of the parameter $F$ than just in a small neighbourhood of the Hopf bifurcation.

The results presented in this paper also illustrate another important point: both qualitative and quantitative aspects of the dynamics of the Lorenz-96 model depend on the parity of $n$. This phenomenon also manifests itself in discretized partial differential equations. For example, for discretisations of Burgers' equation Basto et al. (2006) observed that for odd degrees of freedom the dynamics was confined to an invariant subspace, whereas for even degrees of freedom this was not the case. For the Lorenz-96 model the parity of $n$ also determines the possible symmetries of the model. We will investigate these symmetries,

and their consequences on bifurcation sequences using techniques from equivariant bifurcation theory in forthcoming work (Van Kekem and Sterk, 2017b).

*Code availability.* The scripts used for continuation with AUTO-07p are available upon request from Alef Sterk.

**Appendix A**

**A1    Bounds on the wave number for $F > 0$**

First note that for all $n \geq 4$, with the exception of $n = 7$, there exists at least one integer $j \in [\frac{n}{6}, \frac{n}{4}]$. Indeed, for $n = 4, 5, 6$ this follows by simply taking $j = 1$, and for $n = 8, 9, 10, 11$ this follows by taking $j = 2$. For $n \geq 12$ it follows from the fact that the interval $[\frac{n}{6}, \frac{n}{4}]$ has a width larger than 1 and hence must contain an integer. We now claim that these observations also imply that

$$l^+(n) = \underset{0 < j < n/3}{\arg\max} f(2\pi j/n) \in [\tfrac{n}{6}, \tfrac{n}{4}], \quad n \neq 7.$$

Note that $x \in [\frac{\pi}{3}, \frac{\pi}{2}]$ implies that $f(x) \geq 1$ and $x \in (0, \frac{\pi}{3}) \cup (\frac{\pi}{2}, \frac{2\pi}{3})$ implies $0 < f(x) < 1$. Moreover, $j \in [\frac{n}{6}, \frac{n}{4}]$ implies that $\frac{2\pi j}{n} \in [\frac{\pi}{3}, \frac{\pi}{2}]$. Therefore, $f(2\pi j/n)$ is maximized for some integer $j \in [\frac{n}{6}, \frac{n}{4}]$.

**A2    Asymptotic period for even $n$ and $F < 0$**

Using l'Hopital's $0/0$ rule gives

$$\lim_{x \to \pi/2} (\tfrac{1}{2}\pi - x)\tan(x) = \lim_{x \to \pi/2} \frac{(\tfrac{1}{2}\pi - x)\sin(x)}{\cos(x)}$$
$$= \lim_{x \to \pi/2} \frac{-\sin(x) + (\tfrac{1}{2}\pi - x)\cos(x)}{-\sin(x)}$$
$$= 1.$$





Writing $x = \pi/2 - \pi/2n$ implies

$$\lim_{n \to \infty} \frac{2\pi \tan\left(\frac{\pi}{2} - \frac{\pi}{2n}\right)}{4n} = 1,$$

which in particular implies that

$2\pi \tan\left(\dfrac{\pi}{2} - \dfrac{\pi}{2n}\right) = O(4n).$

### A3   The number of Hopf and double-Hopf bifurcations

The number of Hopf bifurcations of the equilibrium $\boldsymbol{x}_F = (F, \ldots, F)$ for a given dimension $n$ is exactly equal to the number of conjugate eigenvalue pairs which satisfy Theorem 1:

$$N_H = \begin{cases} \lceil \frac{n}{2} \rceil - 1 & \text{if } n \neq 3m, \\ \lceil \frac{n}{2} \rceil - 2 & \text{if } n = 3m, \end{cases} \tag{A1}$$

where we need the ceiling-function if $n$ is odd. Note that if $n$ is a multiple of 3, then $f(\frac{2\pi n}{3}) = g(\frac{2\pi n}{3}) = 0$ which does give a proper complex conjugate pair crossing the imaginary axis and hence the number of Hopf bifurcations has to be decreased by 1.

For the 2-parameter system we can count the number of double-Hopf bifurcations by counting the intersections of the lines in Eq. (16). Since all the lines have a different slope, the number of such intersections is given by

$N_{HH} = \frac{1}{2} N_H (N_H - 1) = \begin{cases} \frac{1}{2}(\lceil \frac{n}{2} \rceil - 1)(\lceil \frac{n}{2} \rceil - 2) & \text{if } n \neq 3m, \\ \frac{1}{2}(\lceil \frac{n}{2} \rceil - 2)(\lceil \frac{n}{2} \rceil - 3) & \text{if } n = 3m, \end{cases}$     (A2)

which shows that the number of double-Hopf points grows quadratically with $n$.

*Author contributions.* Dirk van Kekem performed the research on traveling waves and investigated the dynamics near the double-Hopf bifurcations. Alef Sterk performed the research on standing waves and prepared the manuscript.

*Competing interests.* The authors declare that they have no conflict of interest.



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
