# Peer review of "Wave propagation in the Lorenz-96 model"

_Nonlinear Processes in Geophysics, 2017_

## Referee Comment (RC1) · Anonymous Referee #1 · 22 Nov 2017

This paper presents a detailed analysis of the principal bifurcations in a well known "toy" nonlinear model that has been widely used for the past 20 years as a vehicle for investigating mechanisms for the loss of predictability and onset of spatio-temporal chaos in the atmosphere and oceans. Although this model has been widely used in various contexts, its mathematical properties have not been extensively and systematically studied except in a few specific cases. So one of the distinctive contributions of this manuscript is to apply rigorous analysis tools to study and characterize the first and principal bifurcations of this model as a function of both the model dimensionality n and the forcing parameter F. The results show some clear systematic behavior as n and F are varied, with distinct differences depending on the sign of F. In particular, the present analysis sheds new light on the characteristics of waves that develop within

the Lorenz-96 model and their dependence on n and F. It is good to see the discussion go beyond simply applying numerical tools to obtain diagnostic information on solutions also obtained numerically, but also to examine the structure of the Jacobian, even in fairly high dimensions, to gain mathematical insight as to why certain bifurcations occur.

The paper is generally clear and well written with a reasonable and balanced survey of previous work using the Lorenz-96 model. My only slight concerns are (a) a quibble with the use of the term "standing wave" to describe the stationary wave-like structures found for F<0 and n = 4k+2, and (b) the authors could also make mention of other similar systems exhibiting double-hopf bifurcation to multiple traveling wave solutions (e.g. see Moroz & Holmes 1984. Double Hopf bifurcation and quasi-periodic flow in a model for baroclinic instability, J. Atmos. Sci., 41, 3147-3160). For (a), the term "standing wave" is usually interpreted in physics to mean a time-varying wave-like structure with fixed phase in space only - so can be decomposed into a superposition of two oppositely propagating traveling waves. But the structure described in the paper as "standing" seems actually to be a stationary wave which is effectively a traveling wave of nearly fixed amplitude Doppler-shifted to zero frequency. I would therefore urge the authors to use the term "stationary wave" for these structures in preference to "standing wave", despite precedents elsewhere for the latter (in my view misnomers).

Minor technical points and typos:

P.2 line 15 As well as self-referencing Sterk et al. (2010), you could also mention other earlier studies that also identified Hopf bifurcations associated with the onset of low frequency variability in atmospheres, oceans or laboratory experiments (e.g. Simonnet et al. 2003. Low-Frequency Variability in Shallow-Water Models of the Wind-Driven Ocean Circulation. Part II: Time-Dependent Solutions, JPO, 33, 729-752; Read et al. 1992. Quasi-periodic and chaotic flow regimes in a thermally driven, rotating fluid annulus, JFM, 238, 599-632) P.2 line 21 word missing after "which" - "was"? P.2 lines 24-27 commas recommended after "bifurcation", "that", "F>0", "n", "that" and "F<0", P.3

Fig. 1 Are the braided striations in the Hovmöller plots real or an artifact of the plotting? P.9 line 5 "associated with" not "to" P.22 line 20 Page numbers in reference Frank et al. 2014 seems to be incorrect - should be 1430027?
* * *

---

## Author Comment (AC1) · 14 Dec 2017

First of all, we apologize for the late reply to Referee 1. We would like to thank the Referee for the careful reading of our manuscript and for providing constructive remarks and suggestions. We are happy to read that Referee 1 sees the merits of our work.

The Referee explains that our use of the terminology "standing wave" is not justified. We agree with this, and we thank the Referee for pointing out this misnomer. In the revised paper we will follow the suggestion of the Referee and replace "standing wave" with "stationary wave".

We will add references to papers that identify the Hopf bifurcations associated with the onset of low-frequency variability. We will add the suggested references to Simonnet et

al. and Read et al., but we will also look for additional references. We will add the suggested references on the double-Hopf bifurcation in connection with multiple travelling waves (Moroz & Holmes). On this topic we will also look for additional references.

We thank the Referee for pointing out the typos. We will fix these in the revised manuscript.

The page numbers in the reference to Frank et al. are indeed incorrect. The paper is 14 pages in length, but instead of page numbers we should have included the article number 1430027 in our BibTeX file. We will fix this in the revised manuscript. We thank the Referee for noting this.

Concerning Figure 1. In order to obtain a continuous diagram in the $(j, t)$-plane we have applied linear interpolation between the values $x_j$ and $x_{j+1}$ (see the accompanying caption). The Hovmöller diagram is somewhat "blocky" due to the choice of the time step and the number of linear interpolation points. We will make a figure of higher resolution for the revised manuscript. But perhaps with "braided striations" the Referee means something else. Within the red and blue bands one can see "streaks" of dark red and dark blue, which are indeed artefacts of our linear interpolation procedure. These streaks are precisely located at the $j$-values where $x_j$ is a local maximum or minimum (for fixed values of $t$). At such points the linear interpolation of the $x_j$'s is non-differentiable in $j$, and hence there is a large difference in gradient around either side of such points.

---

## Referee Comment (RC2) · Anonymous Referee #2 · 4 Jan 2018

General comments:

This work analyses the dynamical behaviour, more precisely the bifurcation structure, in the Lorenz96 system as the forcing parameter is varied. For very small forcing, the Lorenz96 system exhibits a stationary state, and the authors focus mainly on the emergence of waves (i.e. periodic solutions) through Hopf (and related) bifurcations from this stationary state. The emerging waves are analysed in terms of their dependence on the dimension of the system. Both the cases of positive and negative forcing are considered and shown to exhibit different behaviour.

Specific comments:

The paper is well written and clearly structured. As the Lorenz96 system is widely used in the atmospheric physics community due to its similarity (in certain aspects) with the climate system, the research question is relevant and timely. I have three major concerns with this manuscript though.

1. The authors fail to draw a connection between their results and the physical phenomena to which the Lorenz96 system pertains, namely atmospheric circulation. A mathematical analysis of the bifurcations in this system would be fine for a mathematics journal, but I believe the general readership of NPG to be ultimately interested in the implications for the physical world. Just to mention a few specific things: the authors should provide a more comprehensive description of the Lorenz96 model and its geophysical interpretation. What is the precise meaning of the model variables; what does the forcing represent physically; (very important!) what is the physical interpretation of *negative* forcing, if any? The properties of the emerging waves should be translated into the physical world, too. What do the results at the end of section 3.1. imply for atmospheric waves; does a larger dimension $n$ mean a finer latitude grid or a bigger planet; what is the interpretation of the limiting wave period?

2. The paper seems to provide very little beyond a previous preprint of the authors (arxiv-preprint 1704.05442). In fact, only the part of the manuscript considering the case of negative forcing seems to contain unpublished material. Given that the authors' results are much less comprehensive in this situation and that this case, so far at least, lacks justification, the question arises whether the manuscript provides enough new material to justify publication in NPG. The authors should provide clear guidance regarding what is new in this manuscript and what is not, and justify why the new material should be of interest to the general NPG readership.

3. The paragraph starting pg.15,l.12 is pretty unclear. It seems to be important though for a major conclusion of this manuscript. The main problem seems to me that you do not define the Neimark–Sacker bifurcation. In fact, the standard definition of the Neimark–Sacker bifurcation pertains to discrete time systems, only. Please clarify. Further, the definition of the "multi–stability lobe" is unclear, and what is the evidence that the two mentioned curves indeed bound such a region?

Technical corrections:

There are some, albeit few and minor, technical issues:

1. Please decide whether you want to write "Figure (No)" or abbreviate as 'Fig. (No)". Same with "Equation".

2. pg.2,l.21, replace "which restricted to" with "which considers only" or similar.

3. pg.2,l.21, replace ", and thereby we" with "; these two manuscripts together" or similar.

4. I think the appropriate technical term is "Jacobi matrix".

5. pg.4,l.2 mention that "equilibrium solution" and "steady flow" mean time–independent solutions.

6. pg.4,l.8 should be "genericity conditions".

7. pg.4,l.9 you never actually define what a Hopf bifurcation is.

---

## Author Comment (AC2) · 8 Jan 2018

We would like to thank the Referee for reading our manuscript and for providing constructive remarks and suggestions.

**Comment 1**

The famous Lorenz-63 model is a Galerkin projection of a fluid dynamical model describing Rayleigh-Bénard convection and hence has a clear physical interpretation. In contrast, the Lorenz-96 model does not have such an interpretation. In fact, Lorenz (2006) writes:

"The physics of the atmosphere is present only to the extent that there

are external forcing and internal dissipation, simulated by the constant and linear terms, while the quadratic terms, simulating advection, together conserve the total energy [...]"

The interpretation of Lorenz is that the variables "[...] may be thought of as values of some atmospheric quantity in $K$ sectors of a latitude circle." (In our paper, we denote the dimension by $n$ instead of $K$.) This is also the interpretation we have adopted in our paper. Since the Lorenz-96 model was not derived from physical principles the sign of the forcing parameter $F$ has no physical interpretation either.

The Hovmöller diagrams of the Lorenz-96 model reveal both travelling waves and stationary waves which have also been observed in physical models, such as the low-order shallow water model studied in Sterk et al. (2010). However, we do not think that it is possible to provide any stronger link with "reality" than that.

It is of course a legitimate question why one would be interested in a model without a physical interpretation. Despite its physical interpretation the Lorenz-63 model has two drawbacks. Firstly, it consists of only three ordinary differential equations. Secondly, for the classical parameter values the model has Lyapunov exponents $0.91$, $0$, and $-14.57$, which makes the model extremely dissipative. Such properties are atypical for atmospheric models. The Lorenz-96 model is a simple nonlinear model with a chaotic regime. Its dimension $n$ can be chosen arbitrarily large, and for suitable values of the parameters the Lyapunov spectrum resembles the spectra found in models obtained from discretized partial differential equations. For these reasons the Lorenz-96 model is used as a test model for a wide range of geophysical applications.

It is an interesting question how the parameter $n$ should be interpreted. It could indeed be interpreted as the resolution of discretization (a larger $n$ implying a finer grid). In fact, some authors interpret the Lorenz-96 model as a discretized partial differential equation; see, for example, Reich & Cotter, "Probabilistic Forecasting and Bayesian Data Assimilation", Cambridge University Press, 2015. However, our results show that

spatiotemporal properties of waves do not always converge as $n \to \infty$. This is also discussed in the context of routes to chaos in our paper arXiv:1704.05442 (v3). Firstly, this makes the interpretation of the Lorenz-96 model as a discretized PDE questionable. Secondly, this implies that the parameters $n$ and $F$ should be chosen carefully when using the Lorenz-96 model for testing purposes. For example, when using the model for experiments in the setting of extreme events, the spatiotemporal properties have a large impact on return times and hence the predictability and statistics of extremes. We will explain this more clearly in the revised manuscript.

The lack of convergence of the dynamics with $n$ could be caused by the fact that the coefficients of all terms in the model are 1. We consider it to be an interesting mathematical problem to investigate whether the coefficients of the model can be changed in such a way that dynamical properties do converge as $n \to \infty$. However, we also feel that this question is beyond the scope of the present manuscript.

**Comment 2**

We agree with the referee that there is some overlap with the present manuscript with our paper arXiv:1704.05442 (v3):

- The eigenvalues of the trivial equilibrium $(F, \ldots, F)$.

- Theorem 1 is a concise summary of Theorems in arXiv:1704.05442 (v3).

- The bounds on the wave number for $F > 0$.

We feel that this overlap is needed in order to (1) give a complete and coherent overview of waves in the Lorenz-96 model and (2) to emphasize the difference between the cases $F > 0$ and $F < 0$. In particular, for $F < 0$ the spatiotemporal properties of waves in the Lorenz-96 model depend on the remainder of $n$ upon division by 4. Moreover, for $F < 0$ the spatial pattern of the wave is determined by the structure of the equilibrium solution that is born through one or two pitchfork bifurcations. To

our knowledge this has not been discussed in the literature before. Also note that in previous work we have not discussed traveling waves for $F < 0$.

The example of the double-Hopf bifurcation for $n = 12$ also appears in arXiv:1704.05442 (v3), but in the current manuscript we show that the number double-Hopf bifurcations grows quadratically with $n$. In addition, we show that for many dimensions of the 2-parameter model there is a pair of double-Hopf bifurcations close to the $F$-axis which causes the co-existence of *three* stable waves. This phenomenon has not been discussed before.

We feel that our results are of interest to NPG readers. Among these readers there are researchers who use the Lorenz-96 model for testing purposes. From Table 1 it becomes clear that many researchers stick with the canonical choices $n = 36$ and $n = 40$. As argued above, such choices can influence the results of numerical experiments. We hope that by giving an coherent overview of the dynamics for various values of $n$ and both $F > 0$ and $F < 0$ users of the Lorenz-96 model will be able to make appropriate parameter choices that suit their purposes.

**Comment 3**

After re-reading this paragraph we agree with the Referee that our explanation is not sufficiently detailed. After a Hopf bifurcation there is a periodic orbit. We can then take a Poincaré section to obtain a discrete-time dynamical system with a stable fixed point. The latter bifurcates through a Neimark-Sacker bifurcation which gives rise to a closed invariant curve, which corresponds to a 2-dimensional torus in the continuous-time system.

In the revised manuscript we will provide a more elaborate explanation of this bifurcation scenario and give additional references. We will also provide a clearer definition of the "multi-stability lobe". The fact that this exists near a Hopf-Hopf bifurcation follows from the normal form theorem of the Hopf-Hopf bifurcation. In general this region does not need to be bounded (but whether it is bounded or not does not affect our results).

We will provide a more detailed explanation in the revised manuscript.

**Technical comments**

Finally, we thank the Referee for suggesting the technical corrections. We will incorporate these in the revision of our manuscript.

---

## Author Response (AR1)

**Wave propagation in the Lorenz–96 model**
**Response to the Referees**

Dirk L. van Kekem        Alef E. Sterk

February 2, 2018

We would like to thank the two Referees for their thoughtful comments and constructive criticism which have helped us to improve our paper. Below we give our point-by-point response to their comments (in blue) and explain how we have incorporated their suggestions in the revised manuscript.

**Response to Referee 1**

1. **[...] a quibble with the use of the term "standing wave" to describe the stationary wave-like structures found for $F < 0$ and $n = 4k + 2$ [...]**

   We agree with the Referee that our use of the terminology "standing wave" is not justified, and we thank the Referee for pointing out this misnomer. In the revised paper we have followed the suggestion of the Referee and replaced "standing wave" with "stationary wave" throughout the text.

2. **[...] the authors could also make mention of other similar systems exhibiting double-Hopf bifurcation to multiple traveling wave solutions [...]**

   In the revised paper we end section 4 with a paragraph in which we give some examples of double-Hopf bifurcations in fluid dynamical models. In addition to the work of Moroz and Holmes (1984) that was suggested by the Referee we have also included the more recent works by Avila (2006), Lewis (2010), Lewis and Nagata (2003, 2005), and Marqués et al. (2002, 2003).

3. **P.2 line 15: As well as self-referencing Sterk at al. (2010), you could also mention other earlier studies that also identified Hopf bifurcations associated with the onset of low-frequency variability in atmospheres, oceans or laboratory experiments [...]**

   In the fourth paragraph of the introduction section we have included references to Simonnet et al. (2003a,b), Te Raa and Dijkstra (2002), Dijkstra et al. (2008), Frankcombe et al. (2009), Read et al. (1992), and Tian et al. (2001).

4. **P.2 line 21: word missing after "which" - "was"?**

   We have rewritten this part as "[...] which considers only the classical case [...]"

5. **P.2 line 24-27: commas recommended after "bifurcation", "that", "$F > 0$", "$n$", "that" and "$F < 0$" [...]**

   This has been corrected as suggested.

6. **P.3 Fig 1 Are the braided striations in the Hovmöller plots real or an artifact of the plotting?**

   In order to obtain a continuous diagram in the $(j, t)$-plane we have applied linear interpolation between the values $x_j$ and $x_{j+1}$ (see the accompanying caption). Within the red and blue bands one can see "streaks" of dark red and dark blue, which are indeed artifacts of our linear interpolation procedure. These streaks are precisely located at the $j$-values where $x_j$ is a local maximum or minimum (for fixed $t$). At such points the linear interpolation of the $x_j$'s is non-differentiable in $j$, and hence there is a large difference in gradient around either side of such points. Alternatively, one could use an interpolation method based on smooth functions, but we do not feel that this would add much to our plots.

7. **P.9 line 5 "associated with" not "to"**

   This has been corrected as suggested.

8. **P.22 line 20 Page numbers in reference Frank et al. 2014 seems to be incorrect [...]**

   The page numbers in the reference to Frank et al. (2014) were indeed incorrect. The paper is 14 pages in length, but instead of page numbers we should have included the article number 1430027 in our BibTeX file. We have corrected this, and we thank the Referee for noting the mistake.

**Response to Referee 2**

1. **The authors should provide a more comprehensive description of the Lorenz96 model and its geophysical interpretation.**

   The famous Lorenz-63 model is a Galerkin projection of a fluid dynamical model describing Rayleigh-Bénard convection and hence has a clear physical interpretation. In contrast, the Lorenz-96 model does not have such an interpretation. In fact, Lorenz (2006) writes:

   > "The physics of the atmosphere is present only to the extent that there are external forcing and internal dissipation, simulated by the constant and linear terms, while the quadratic terms, simulating advection, together conserve the total energy [...]"

   The interpretation of Lorenz is that the variables "[...] may be thought of as values of some atmospheric quantity in $K$ [$n$ in our paper] sectors of a latitude circle." This is also the interpretation we have adopted in our paper. In the paper "Regimes in simple systems" (Journal of the Atmospheric Sciences, 63, 2006, pp. 2056–2073) Lorenz writes:

> "The variations of $X_n$ [$x_j$ in our paper] are intended to simulate the behavior of some atmospheric quantity at $N$ [$n$ in our paper] equally spaced grid points about a latitude circle, but to my knowledge, the system cannot be obtained by truncating any realistic atmospheric model."

Since the Lorenz-96 model was not derived from physical principles the sign of the forcing parameter $F$ has no physical interpretation either. The Hovmöller diagrams of the Lorenz-96 model reveal both travelling waves and stationary waves which have also been observed in physical models, such as the low-order shallow water model studied in Sterk et al. (2010). However, we do not think that it is possible to provide any stronger link with "reality" than that.

After the first paragraph in the introduction section we included a new paragraph which aims to clarify why the the Lorenz-96 model is useful despite the lack of a clear physical interpretation.

**Does a larger dimension $n$ mean a finer latitude grid or a bigger planet; what is the interpretation of the limiting wave period?**

The parameter $n$ could indeed be interpreted as the resolution of discretization with a larger $n$ implying a finer grid. In fact, some authors interpret the Lorenz-96 model as a discretized partial differential equation; see, for example, Reich & Cotter, "Probabilistic Forecasting and Bayesian Data Assimilation", Cambridge University Press, 2015. However, our results show that spatiotemporal properties of waves do not always converge as $n \to \infty$. The lack of convergence of the dynamics with $n$ could be caused by the fact that the coefficients of all terms in the model are 1. We consider it to be an interesting mathematical problem to investigate whether the coefficients of the model can be changed in such a way that dynamical properties do converge as $n \to \infty$. However, we also feel that this question is beyond the scope of the present manuscript.

Our results imply that the parameters $n$ and $F$ should be chosen carefully when using the Lorenz-96 model for testing purposes. In Sterk and Van Kekem (2017) we have shown that the predictability of extreme events in the Lorenz-96 model strongly depends on $n$ and $F$. In the revised manuscript we have made this more explicit.

2. **The paper seems to provide very little beyond a previous preprint of the authors (arxiv-preprint 1704.05442). In fact, only part of the manuscript considering negative forcing seems to contain unpublished material. Given that the authors' results are much less comprehensive in this situation and that this case, so far, lacks justification, the question arises whether the manuscript provides enough material to justify publication in NPG. The authors should provide clear guidance regarding what is new in this manuscript and what is not, and justify why the new material should be of interest to the general NPG readership.**

Note that there are three versions of the manuscript arXiv:1704.05442. The latest version (version 3, hereafter referred to as V3) dates from December 20, 2017 and will be published in Physica D. We will remove versions 1 and 2 in due time in order to prevent confusion.

V3 is aimed at a mathematical audience and contains mostly technical details, including proofs, normal form computations, and routes to chaos. In contrast, the present manuscript is aimed at researchers who use the Lorenz-96 model in their work. We agree with the Referee that there is some overlap with the present manuscript and V3, namely:

- the eigenvalues of the equilibrium $(F, \ldots, F)$;

- theorem 1, which is a concise summary of theorems in V3;

- the bounds on the wave number for $F > 0$ and the limiting wave period.

However, we feel that this overlap is needed in order to (1) give a complete and coherent overview of waves in the Lorenz-96 model and (2) to emphasize the difference between the cases $F > 0$ and $F < 0$. In particular, for $F < 0$ the spatiotemporal properties of waves in the Lorenz-96 model depend on the remainder of $n$ upon division by 4. Moreover, for $F < 0$ the spatial pattern of the wave is determined by the structure of the equilibrium solution that is born through one or two pitchfork bifurcations. To our knowledge this has not been discussed in the literature before. Also note that in previous work we have not discussed traveling waves for $F < 0$.

We are not sure what the Referee means with the remark that our results are "less comprehensive" for $F < 0$. Does it mean a lack of rigorous proof? For $F > 0$ and $F < 0$ with $n$ odd we were able to analytically compute the spatiotemporal properties of the waves. However, our example for $n = 4$ shows that this is not always possible.

The double-Hopf bifurcation for $n = 12$ also appears in V3, but the discussion in V3 is technical and focuses on detailed normal form computations. In the current manuscript we show that the number double-Hopf bifurcations grows quadratically with $n$. In addition, we show that for many dimensions of the 2-parameter model there is a pair of double-Hopf bifurcations close to the $F$-axis which causes the co-existence of *three* stable waves. This phenomenon has not been discussed in V3.

We believe that our manuscript is sufficiently interesting to the readers of NPG. Among the readers of NPG there are researchers who use the Lorenz-96 model in their work. For example, see the references Basnarkov and Kocarev (2012), Lucarini and Sarno (2011), Orrell et al. (2003), Sterk et al. (2012), and Trevisan and Palatella (2011). Table 1 shows that many researchers stick with the canonical choices $n = 36$ and $n = 40$. As argued above, these choices can influence the results of numerical experiments. We hope that by giving an coherent overview of the dynamics for various values of $n$ and both $F > 0$ and $F < 0$ users of the Lorenz-96 model will be able to make appropriate parameter choices that suit their purposes.

3. **The paragraph starting pg.15, l.2 is pretty unclear. It seems to be important though for a major conclusion of the manuscript. The main problem seems to me that you do not define the Neimark-Sacker bifurcation. In fact, the standard definition of the Neimark-Sacker bifurcation pertains to discrete-time systems only. Please clarify. Further, the definition of "multi-stability lobe" is unclear, and what is the evidence that the two mentioned curves indeed bound such a region?**

We fully agree with the Referee that our explanation was not sufficiently detailed.

In the revised manuscript we explain what Neĭmark-Sacker bifurcations in discrete-time and continuous-time systems are. We also explain how these bifurcations lead to coexistence of periodic orbits. To that end we describe the successive bifurcations that occur while keeping $G = 0.1$ fixed and letting $F$ increase. This scenario is not limited to the Lorenz-96 model. In fact, it occurs in any system with a double-Hopf bifurcation of type I, which follows from the normal form analysis in Kuznetsov (2004).

4. **Please decide whether you want to write "Figure (No)" or abbreviate as 'Fig. (No)'.**

The section *Manuscript preparation guidelines for authors* on the website of NPG explicitly states the following:

> The abbreviation "Fig." should be used when it appears in running text and should be followed by a number unless it comes at the beginning of a sentence, e.g.: "The results are depicted in Fig. 5. Figure 9 reveals that...".

We have checked carefully that we have followed this guideline in a consistent manner.

5. **pg.2, l.21, replace "which restricted to" with "which considers only" or similar.**

We have rewritten this part as "[...] which considers only the classical case [...]"

6. **pg.2, l.21, replace and "thereby we" with "; these two manuscripts together" or similar.**

We have rewritten this part as "; these two manuscripts together give a comprehensive picture of wave propagation in the Lorenz-96 model."

7. **I think the appropriate technical term is Jacobi matrix.**

In the languages like German or Dutch the $m \times n$ matrix with first-order partial derivatives of a map $F : \mathbb{R}^n \to \mathbb{R}^m$ is referred to as "Jacobi matrix". However, we have consulted various English text books on mathematics and in those texts the term "Jacobian matrix" is used. Hence, we have not changed the terminology.

8. **pg.4, l.2 mention that "equilibrium solution" and "steady flow" mean time-independent solutions.**

We have rewritten this sentence as follows:

> "Assume that for the parameter value $\mu_0$ the system has an equilibrium $\vec{x}_0$; this means that $\vec{f}(\vec{x}_0, \mu_0) = \vec{0}$ and hence $\vec{x_0}$ is a time-independent solution of eq. (2)."

9. **pg.4, l.8 should be "genericity conditions".**

The word "nongenericity conditions" was indeed a slip of the pen. We have changed it into "non-degeneracy conditions" which is the appropriate technical term in the literature on bifurcation theory.

10. **pg.4, l.9 you never actually define what a Hopf bifurcation is.**

We agree with the Referee that the flow of this paragraph is not completely logical. This indeed leads to using the word "Hopf bifurcation" without explaining it.

In the revised manuscript we have rewritten the starting sentences of this paragraph as follows:

[revised manuscript text omitted]

---

## Author Response (AR2)

**Wave propagation in the Lorenz–96 model**

**Final response**

Dirk L. van Kekem         Alef E. Sterk

April 11, 2018

We would like to thank all the Referees for their thoughtful comments and constructive criticism which have helped us to improve our paper. We are happy to read that our paper is now acceptable for publication.

We decided to refer to *all* software packages that we have used during our research. Packages like AUTO–07P and MatCont are developed by researchers and free of charge to use. Therefore, we feel that some form of credit by means of a citation is justified. Other packages, such as Mathematica, are commercial, but we feel that it would not be appropriate to make a distinction between free and commercial software in our references. Hence, we have also decided to keep the reference to Mathematica, but we did remove the extraneous "2016" and corrected the capitalization.